# Treatment of Advanced Basal Cell Carcinoma with Hedgehog Pathway Inhibitors: A Multidisciplinary Expert Meeting

**DOI:** 10.3390/cancers13225706

**Published:** 2021-11-15

**Authors:** Vincenzo De Giorgi, Federica Scarfì, Luciana Trane, Flavia Silvestri, Federico Venturi, Biancamaria Zuccaro, Giuseppe Spinelli, Silvia Scoccianti, Francesco De Rosa, Emi Dika, Caterina Longo

**Affiliations:** 1Section of Dermatology, Department of Health Sciences, University of Florence, 50100 Florence, Italy; scarfif@gmail.com (F.S.); flavia.silvestri25@gmail.com (F.S.); federico.venturi@unifi.it (F.V.); biancamaria.zuccaro@unifi.it (B.Z.); 2Cancer Research “AttiliaPofferi” Foundation, 50100 Pistoia, Italy; luciana.trane@gmail.com; 3Maxillofacial Surgery Department, Azienda Ospedaliero-Universitaria Careggi, 50100 Florence, Italy; info@giuseppespinelli.it; 4Radiation Oncology Unit, Azienda Toscana Centro, 50100 Florence, Italy; silvia.scoccianti@uslcentro.toscana.it; 5Immunotherapy-Cell Therapy and Biobank, Istituto Scientifico Romagnolo per lo Studio e la Cura dei Tumori (IRST) “Dino Amadori” IRCCS, 47014 Meldola, Italy; francesco.derosa@irst.emr.it; 6Dermatology, Department of Experimental, Diagnostic and Specialty Medicine, University of Bologna, 40138 Bologna, Italy; emi.dika3@unibo.it; 7Dermatology, IRCCS Policlinico di Sant’Orsola, Via Massarenti 9, 40138 Bologna, Italy; 8Department of Dermatology, University of Modena and Reggio Emilia, 42100 Modena, Italy; caterina.longo@unimore.it; 9Azienda Unità Sanitaria Locale—IRCCS di Reggio Emilia, Centro Oncologico ad Alta Tecnologia Diagnostica-Dermatologia, 41121 Reggio Emilia, Italy

**Keywords:** skin cancer, vismodegib, sonidegib, therapy, cemiplimab

## Abstract

Despite recent progress and the publishing of several clinical guidelines on the management of advanced basal cell carcinoma, there is still no comprehensive set of clinical guidelines addressing the complexity inherent to the use of Hedgehog pathway inhibitors in the treatment of advanced basal cell carcinoma in real-world clinical practice. To develop practical and valuable tools that help specialists improve the clinical management of these patients, we sought the opinion of expert physicians with extensive knowledge and experience in the treatment of advanced basal cell carcinoma.

## 1. Introduction

Basal cell carcinoma (BCC) accounts for almost 80% of all nonmelanoma skin cancer cases, with 2–3 million BCC cases per year worldwide [1,2]. The majority of cases can be cured via complete surgical excision as recommended by international guidelines [3,4]. In some cases, radiotherapy (RT) proves to be a valid therapeutic aid, especially in older patients. Although BCC is often underestimated in terms of aggressiveness, surgery cannot be indicated in the advanced stages of the disease, especially in the locally advanced (laBCC) and metastatic disease phases. For these complex scenarios, two systemic drugs belonging to a group of molecules known as Hedgehog pathway inhibitors(Hhis) are suggested: vismodegib and sonidegib [3,4,5]. In the last few decades, vismodegib and sonidegib, which act as suppressors of the transmembrane protein Smoothened, have been approved in the USA, Europe, Switzerland, and Australia with different indications [6,7,8,9,10,11,12,13]. Sonidegib and vismodegib are increasingly being used as a valid medical therapy in the management of laBCC, when surgery and/or RT are contraindicated or inappropriate for the patient. Several studies support the effectiveness and safety of these drugs. Regarding the efficacy of Hhis as a treatment for laBCC, we only discuss the BOLT study for sonidegib and the ERIVANCE study for vismodegib as they are pivotal trials with comparable design and endpoints. The ERIVANCE nonrandomised, open-label study was designed to register vismodegib at a dose of 150 mg/day. In comparison, the BOLT study was a randomised, double-blind trial that evaluated two different doses of sonidegib (200 and 800 mg/day); this study was ultimately used to register sonidegib at 200 mg/day, the dose which showed the best balance between efficacy and safety. In terms of efficacy, the objective response rate of sonidegib was 56.1% according to the mRECIST response (central review) [14]. The objective response rate for vismodegib was 47%. The duration of response for sonidegib was 26 months. In the ERIVANCE study, the duration of response for vismodegib was 9.5 months, although the two drugs were not directly compared. However, these two drugs have different pharmacokinetics profiles. Vismodegib has a volume of distribution of 16–27 L, suggesting that it is largely confined to the plasma and has limited tissue penetration. In contrast, sonidegib seems to be more lipophilic than vismodegib and has a volume of distribution of >9000 L, indicating extensive distribution in the tissues [14]. The concentration of sonidegib is provided to be six times higher in the skin than in plasma; on the other hand, the cutaneous concentration of vismodegib was not measured. Regarding treatment schemes, sonidegib is the only Hhi with an approved alternate-day dose.

The optimal management of these drugs is still a topic of debate among clinicians belonging to different specialties, such as dermatologists, oncologists, surgeons, and radiotherapists. There is still no multidisciplinary consensus on topics such as therapeutic schemes, patient choice, the duration of therapy, and the use of Hhis in neoadjuvant therapy. In daily practice, clinicians encounter complex scenarios that are not always described in the treatment guidelines, and clear recommendations are lacking. Furthermore, even rigorous adherence to protocols developed with the best knowledge of these drugs and their effects on larger sample sizes is not always productive in real-life situations. Here we report the outcomes of a multidisciplinary expert meeting, held in Florence, Italy, in October 2020, on decision making regarding the use of Hhis in complex clinical and managerial situations that arise during the treatment of advanced BCC. This paper summarises the topics discussed and formulates recommendations for future decision-making.

## 2. Materials and Methods

On 10 October 2020, a panel of experts was convened at the University of Florence in Florence, Italy. The expert meeting consisted of seven Italian skin cancer specialists: dermatologists, oncologists, surgeons, and radiotherapists invited according to personal title and specific field of expertise. The goal was to reach a consensus on key aspects of the optimal management of advanced basal cell carcinoma patients, especially those with laBCC, based on evidence and the experts’ clinical experience. The discussion was carried out in three stages: (1) a round of online surveys to establish the main topics and formulate the questions to be discussed, (2) a face-to-face meeting to discuss the main topics and reach a consensus for each of them, and (3) an analysis of the results and a concluding discussion via online meeting.

The panel of experts compiled a list of relevant clinical topics to define the optimal management of patients with laBCC. The following questions were analysed:What criteria should be used to diagnose patients with laBCC, and which therapy is most appropriate for these patients?How can clinicians manage the adverse side effects of Hhis and improve patient adherence to Hhi treatment plans?When can immunotherapy be used as a second-line treatment after Hhis?What are the educational and research needs regarding laBCC treatment with Hhis?

## 3. Results

A consensus was achieved among the experts for the selected questions. 

### 3.1. Question 1

What criteria should be used to diagnose patients with laBCC, and which therapy is most appropriate for these patients?

#### 3.1.1. Background

The definition of “advanced” BCC was introduced at the beginning of the Hhi era. The recognition of BCC as advanced depends on multiple parameters, such as tumour location (e.g., critical zones such as the face), the occurrence of bone infiltration, tumour size >2 cm, depth of the lesions, and tumour history (e.g., growth speed, relapse time, and the number of lesions). Recently, Dummer et al. [14] have suggested some additional criteria to help identify laBCC patients who are appropriate candidates for Hhi treatment: the presence of more than five BCCs in patients with a genetic syndrome and the relapse of lesions over 10 mm in diameter in a critical location even after two surgeries. 

However, any BCC that is “difficult to treat” can generally be considered laBCC according to European consensus-based interdisciplinary guidelines [4]. This implies that the psychological impact of the condition and the patient’s social context should also be considered during diagnosis. For instance, many laBCC cases occur in neglected people who cannot take care of themselves. Moreover, both European consensus-based interdisciplinary guidelines [4] and several other sets of guidelines [13] have highlighted the importance of the patient’s choice and opinion regarding laBCC treatment. The most important step in the treatment of laBCC is to identify the best approach for each patient as soon as possible. The operability of BCC must be evaluated in a multidisciplinary manner whereby each specialist employs their own specific skills. In cases with a limited number of lesions (i.e., 1–2), it is not difficult to establish an effective treatment plan according to the localization of the tumour. Indeed, if neglected, BCC in the periorificial areas of the face, especially the periocular region or the nose, can lead to severe consequences [15]. Surgery is generally contraindicated when it would cause severe deformities and when lesions relapse after multiple surgeries [16]. When surgery or RT is contraindicated, pharmacological treatment with Hhis is the next logical option. Vismodegib, the first Hhi to be developed, is orally administered at a daily dose of 150 mg, and it is indicated for both laBCC and metastatic BCC patients. Sonidegib, which was developed after vismodegib, is also an orally administered drug. The daily dose of sonidegib is 200 mg, and it is indicated exclusively for laBCC patients.

Conforming BOLT results using RECIST-like criteria resulted in patients with laBCC treated with 200-mg sonidegib showing a slightly higher overall response rate (60 vs. 47.6%) than those in therapy with vismodegib. In addition, sonidegib had an approximately 10% lower incidences of most AEs compared with vismodegib, and the time to onset of AEs also indicated that patients treated with sonidegib might experience AEs slightly later than with vismodegib [17,18,19,20]. Another significant advantage of using sonidegib is the possibility of an alternate-day dosing schedule in order to manage better the AEs [21].

#### 3.1.2. Discussion and Consensus for Question 1

A surgeon’s concept of operability should be established based on the best techniques available at their surgical centre. Surgery should be considered if it will yield the radical excision of the tumour mass. According to the panel of experts, even if a lesion is operable, medical therapy with Hhis should be considered if radical surgical removal of the tumour is unlikely. The choice between surgical intervention and the therapy target option is also affected by the risk of leaving patients with disfigurements and the difficulty of planning future rescue surgeries in case of disease progression. Therefore, surgery must also be sustainable from the patient’s psychological point of view; surgeons must consider the health-related quality of life after surgery. In the opinion of the panellists, when the patient refuses surgical or RT treatment for psychological reasons, the operable tumour becomes de facto inoperable.

The outcome of RT treatment largely depends on the radiotherapist’s expertise and the available technological apparatus, which can help achieve an effective superficial dose. It is crucial to consider the localisation of the la BCC. RT is contraindicated when the tumour is located in a risky area, such as areas in close proximity to delicate organs (e.g., the eyelids); areas of irregular thickness (e.g., the nose); and areas where the surface is not flat, making it difficult to provide a homogeneous dose.

On the one hand, it is undeniable that technology’s rapid development has helped clinicians treat unwieldy tumours. On the other hand, it is still challenging to define reproducible criteria to identify laBCC and decide the best treatment for these tumours. In any case, the choice of treatment should always be considered in a multidisciplinary fashion, and the management of laBCC must be tailored to each patient. In the opinion of the panellists, even if laBCC is diagnosed by dermatologists in 90% of cases, the pathway to the diagnosis of these patients must always start with a multidisciplinary assessment to avoid errors. When planning the management of laBCC patients, clinicians must also consider future risks, including the loss of efficacy of Hhis. Loss of efficacy of Hhis can be a relevant issue for younger patients. Therefore, therapy outcomes with Hhis must be planned for the medium-long term, and the panellists think that their use in the neoadjuvant setting can be highly beneficial. In fact, the panellists believe that in certain situations, such as tumours in the periocular region, treatment with Hhis before surgery and/or RT may be useful to reduce the mass of the tumour and render a previously untreatable lesion treatable. Recently a multicenter, open-label, phase 2 trial (VISMONEO study) confirmed the crucial role of vismodegib in the neoadjuvant setting. In particular, neoadjuvant vismodegib allows for a downstaging of the surgical procedure for laBCCs in functionally sensitive locations of the face [22].

The laBCC tumors of patients with Gorlin–Goltz syndrome must be considered inoperable and must be treated with Hhi therapy, even if the patient has lesions that can be treated individually with surgery. According to the clinical experience of the panellists, Hhi treatment usually leads to an optimal response in patients with Gorlin–Goltz syndrome. Indeed, these patients respond better to Hhi treatment, achieve a longer duration of response, and experience a higher success rate with rechallenge therapy than patients without Gorlin–Goltz syndrome. Despite these, given the genetic basis of the disease, unavoidably BCCs recur after drug interruption. Although, as observed by Herms et al. [23], patients with or without Gorlin–Goltz syndrome who achieved complete response (CR) and stopped vismodegib, have significant long terms responses and, after relapses, are still responsive to vismodegib treatment. In conclusion, the panellists believe that the importance of Hhis as adjuvants or neoadjuvants in the treatment of laBCC should be highlighted.

### 3.2. Question 2

How can clinicians manage the adverse side effects of Hhis and improve patient adherence to Hhi treatment plans?

#### 3.2.1. Background

A high percentage of patients undergoing treatment with vismodegib or sonidegib experience at least one adverse event (AE). The most common AEs, considering this term the sum of grade ≤2 and ≥3 AEs, include muscle spasms (54% of patients taking sonidegib and 71% of patients taking vismodegib), alopecia (49% with sonidegib and 66% with vismodegib), and dysgeusia (44% with sonidegib and 56% with vismodegib) [14,20,21,24,25].

Based on the available data regarding BOLT and ERIVANCE studies, sonidegib seems to have a lower incidence of AEs, and the AEs related to the study treatment resulted in less frequent and less severe compared to vismodegib [14,24].

Muscle spasms, the most common AE, are associated with a high level of creatine kinase; it has been hypothesised that Hhis can act as agonists of the noncanonical Hedgehog pathway, inducing calcium channel activation and thus leading to the observed spasms [26].

Alopecia is an AE observed during both vismodegib and sonidegib treatment. The molecular mechanism underlying this AE is an arrest in the transition from the telogen to the anagen phase of hair follicle growth; the Hedgehog pathway mediates this transition [27].

Dysgeusia in tandem with weight loss is also a noteworthy AE that occurs during Hhi therapy. Reductions in weight, most of which were grade ≥3, were observed in 5% and 9% of patients treated with sonidegib [1] and vismodegib, [2] respectively. Therefore, early nutritional screening and subsequent nutritional management are recommended to avoid malnutrition among patients.

Quality of life outcomes are an important parameter to assess treatment efficacy.

Two tools (EORTC QLQ-C30 and EORTC H&N35) were used in the BOLT trials, demonstrating that quality of life was maintained or improved among treated patients. Patients with laBCC treated with vismodegib showed no positive changes from baseline on either the physical or emotional portions of the Short Form-36 questionnaire in ERIVANCE. However, these data cannot be compared, as different health-related QoL scales were used, and the assessment frequency also differed. A review of the QoL outcomes from the STEVIE study showed that vismodegib was associated with clinically meaningful improvement in the emotional domain using the Skindex-16 scale and MD Anderson Symptom Inventory questionnaires [28].

An analysis of safety by age group was performed in the ERIVANCE study, with a median drug exposure of 9 months in patients aged ≥65 years and 10 months in patients ≤65 years old. Grade ≥ 3 AEs were most commonly found in older patients. Nevertheless, the clinical activity of the drug was similar regardless of age. Based on the data presented in the BOLT and ERIVANCE studies, sonidegib seems to have a lower incidence of AEs overall, and the AEs related to the study treatment were less frequent and less severe compared with vismodegib. Moreover, the data are consistent with a previous analysis of safety at a 42-month follow-up [14,23,24]. As already mentioned, the cutaneous concentration of sonidegib is six times higher than that of theplasmaand Theoretically, due to the differences in the pharmacokinetics, Sonidegib is more likely to be distributed in the skin than vismodegib, which may explain potential differences between the two molecules regarding efficacy and toxicity. For the management of AEs, treatment with sonidegib also offers the option for dose modification: an initial dose of 200 mg daily can be reduced to 200 mg every other day. If the same AEs also occur with alternate daily dosing, an interruption in treatment is required. Sonidegib can be resumed after the resolution of AEs to grade ≤ 1. In the BOLT studies, patients requiring dose reductions continued to benefit from sonidegib treatment.

#### 3.2.2. Discussion and Consensus for Question 2

Treatment with sonidegib results in 10–15% fewer detectable AEs and has a more optimal efficacy when compared with vismodegib. Additionally, treatment with sonidegib allows for excellent manageability of AEs with the possibility of alternate-day dosing and dosage management. However, a challenging topic remains in the management of patients that achieve the complete clearance of a lesion: based on the opinion of the experts on the panel, if the patients tolerate the drug, there is no indication to interrupt the treatment. Over time, it may be useful to switch to pulsed therapy for as much as one week a month, considering the ideal pharmacokinetics of sonidegib. However, treatment should be discontinued if the patients show tolerability issues. Moreover, clinical cases with vismodegib showed that when patients’ disease progression restart the treatment after suspension, they still respond to the treatment [17,18]. Furthermore, in the face of disease progression after treatment with vismodegib, a new treatment protocol with sonidegib may be valuable and vice versa.

Another challenging issue is represented by the persistence of AEs after the interruption of treatment with Hhis. In the literature and according to the experience of the panellists, alopecia cases are reported even after a two-year suspension of vismodegib, and muscle spasms one year after the treatment’s suspension. Sonidegib seems to have a more favourable safety profile than vismodegib, especially concerning muscle spasms, dysgeusia, and alopecia. Nonetheless, the onset and the prolongation of such AEs represent a critical issue for patient management as the treatment must be performed in the long term. All the panellists agree that the side effects of Hhi therapy must be fully explained to the patient. The use of detailed, informative material focused on the management of AEs and dedicated to patients and their general practitioners can be useful to achieve adherence to the treatment plan. In fact, it is the common experience of the panel that informed and motivated patients better tolerate side effects. Moreover, a patient’s diary for monitoring the onset and course of any AEs can provide additional helpful information for AE management.

About the management of AEs, the panellist discussed the main recommendations. Muscle spasms are more frequent with vismodegib. The concurrent use of other drugs or supplements (magnesium, etc.) has returned mixed results that are perhaps more related to a placebo effect on the patient. Some patients benefit somewhat from the use of central myorelaxant drugs. The use of calcium antagonists is not effective [22] and not advisable in patients with comorbidities. The panellists agree that the best management of this AE is represented by physical activity in general and physical exercises (walking, stretching, etc.); this is most effective if done immediately before going to sleep. As for ageusia and the consequent weight loss, especially in elderly patients who are often already debilitated, all the panellists agree on sending the patient to a dietician before the start of the treatment itself and not when the problem becomes evident during treatment. This foresight facilitates patient compliance with the treatment.

Lastly, the panellists agree that alopecia has a smaller impact on patient compliance than the other AEs (see ageusia and muscle spasms), and among the various effects it is the most tolerated. The experts agree that the benefit of minoxidil use on Hhi-triggered alopecia is limited.

Patients with Gorlin–Goltz syndrome respond favourably to Hhi treatment. Due to their experience of multiple, often disfiguring and invasive surgeries, they usually tolerate AEs better than patients without this syndrome. Vismodegib has demonstrated the capacity to reduce BCC tumour burden and stop the growth of new BCCs in patients with Gorlin–Goltz syndrome, although close to half of patients discontinued the treatment due to AEs. Sonidegib has shown promising efficacy, exhibiting partial or complete clinical clearance of target BCCs, decreased tumour burden in all patients, and histological clearance in more than half of the cases. Moreover, sonidegib appears to be well-tolerated.

### 3.3. Question 3

When immunotherapy can be used as a second-line treatment after Hhis?

#### 3.3.1. Background

Thus far, when tumor progresses under Hhi therapy the only remaining therapy is immune checkpoint blocking antibodies, [29] and cemiplimab, the programmed cell death 1 protein (PD-1) monoclonal antibody, is the only one approved in 2021 by the FDA and EMA for the treatment of locally advanced or metastatic BCC.

Other immunotherapies are in clinical trials [30,31,32,33]. The immunogenicity of BCC has been highlighted in recent articles. For example, the pathogenesis of BCC was shown to be closely related to a large tumor mutational burden induced by ultraviolet (UV) [34] radiation, which causes damage to cellular deoxyribonucleic acid (DNA), leading to a great number of tumoral neoantigens and subsequently, a certain degree of immunogenicity. However, even on this foundation, BCC has shown low immunogenicity, a phenomenon that can be attributed to the downregulation of antigenic presentation. Moreover, in BCC, there is an increase in regulatory T lymphocytes and an immunosuppressive effect dictated by the presence of interleukin 10 and T-helper(Th)2 cytokines [35,36]. This network can lead to a tumor immune escape.

A recent study has shown how the use of Hhis can increase the immunogenicity of BCC, rendering the neoplasm more susceptible to anti-PD-1 therapy [37]. In fact, tumor regression induced by Hhis can be followed by an alteration in the microenvironment dictated by the fall of immunological privilege. This can occur through a local increase in cytotoxic T lymphocytes and activation of the adaptive immune response with a change in the cytokine milieu and upregulation of major histocompatibility complex (MHC) class I molecules. This could explain how patients with locally advanced or metastatic BCC who underwent Hhi therapy followed by immunotherapy with cemiplimab achieved good therapeutic responses [38].

Research is also moving toward the use of novel immune modulatory vaccines in skin tumors. A recent Phase IIa study of the role of peptide vaccine IO103, which targets the major PD-1 ligand (PD-L1), showed a partial response in target tumors and a 12.8% reduction in tumor size [39].

To date, a few reports have described lasting partial or complete clinical responses to immunotherapy in people with BCC; some have not been so lucky. The cause of treatment resistance remains to be elucidated. A case report illustrated how new skin BCC appeared in a patient with metastatic BCC during treatment with nivolumab, leading to a hypothesis that the ideal targets of immunotherapy are those with BCC and a large tumor mutational burden [40]. In other words, it would be necessary to change how BCC therapy is approached: this type of treatment is more effective in the most advanced stages of the disease rather than in the early stages. Moreover, much remains to be clarified about the predictive factors of response to immunotherapy.

#### 3.3.2. Discussion and Consensus for Question 3

The panel of experts agrees that studies identifying the actual effectiveness of immunotherapy as a treatment for local advanced and/or metastatic BCC must be developed. However, at present, they believe that Hhis therapy remains the first-choice treatment for advanced BCC. Moreover, the use of the monoclonal antibody cemiplimab should be used only for patients that develop resistance under Hhi therapy rather than patients who don’t tolerate Hhi where experienced physicians can avail the AE management strategies available such as dose interruptions, on label dose reduction for Sonidegib, and supportive medications. Based on personal experience, experts believe that it could be useful to continue Hhis therapy in those who do not respond to it by associating it instead with monoclonal antibodies (e.g., cemiplimab), given the evidence of greater immunogenicity of BCC during Hhis therapy [37], rendering the neoplasm more susceptible to anti-PD-1 therapy. Furthermore, this combination therapy could also solve the problem with basal-squamous carcinoma, which has metastatic potential in both the basal and squamous components.

### 3.4. Question 4

What are the educational and research needs regarding laBCC treatment with Hhis?

#### Discussion and Consensus for Question 4

All the experts on the panel agree that one of the most burning questions in Hhi treatment is how long the treatment should be continued after the tumour shrinks or resolves. Unfortunately, at present there are no well-constructed studies in the literature that address this topic. For these reasons, a phase II study examining various therapeutic regimens should be conducted in patients who achieve a complete response; ideally, this study would be accompanied by further studies comparing prolonged and intermittent Hhi suspension.

A more specific study addressing suspension versus intermittent treatments should be performed in patients with Gorlin–Goltz syndrome. Due to the absence of such studies at the present moment, the criteria for Hhi suspension are empirical. The experts’ most common approach consists of suspending treatment when the patient achieves complete remission and restarting it at the time of disease progression. The particular pharmacokinetic qualities of sonidegib, such as its half-life of 28 days, allow for the use of this drug in a pulsed therapy protocol, minimising side effects and maximising lasting clinical benefits.

Another topic that requires further exploration is the use of Hhis for adjuvant and neoadjuvant therapies. The use of Hhis in adjuvant or neoadjuvant treatment is not allowed in numerous countries due to the lack of extended studies. The availability of publications can facilitate communication with healthcare authorities. Even a pilot study could help authorise the use of such drugs outside standard guidelines. Neoadjuvant and adjuvant approaches to Hhi use are discussed in dedicated tumour board meetings, and such therapies could be necessary when radical surgery is not possible or to make a previously inoperable tumour operable. These courses of action are potentially helpful in treating tumours localised to particular anatomical areas, such as the periocular region or the face in general. The characteristics of sonidegib make it a particularly attractive candidate for use in neoadjuvant therapy; however, the decision to use it should be considered on a case-by-case basis since data on this indication are sparse. There is limited evidence regarding surgical margins after neoadjuvant therapy with Hhis, and no data are proving that adjuvant therapy reduces the risk of recurrence after surgical intervention.

The experts also agree on the need for educational activities directed towards the various specialists performing laBCC therapies. An example of a potentially valuable activity is an educational program for general practitioners, geriatric physicians, and surgeons working at retirement homes or clinics that provides awareness about early detection or diagnosis of laBCC. Such educational courses should be organised through local medical organisations, such as local health administrations.

## 4. Conclusions

The main take-home messages of this expert meeting can be summarized, for each question, in the following key points.


**Question 1: What criteria should be used to diagnose patients with laBCC, and which therapy is most appropriate for these patients?**



**Key Points**
The management of each case of laBCC must always be conducted in a multidisciplinary manner and should be tailored to each patient.Even if a lesion is operable, medical therapy with Hhi should be considered if radical surgical removal of the tumour is unlikely.Surgery must also be sustainable from the psychological point of view of the patient; surgeons must also consider the aspects of health-related quality of life after surgery.When a patient refuses surgical treatment for psychological reasons, an operable tumour becomes de facto inoperable.Patients with Gorlin syndrome are considered not amenable to surgery due to many lesions, and Hhi treatment in these patients usually lead to a favourable response.



**Question 2: How can clinicians manage the adverse side effects of Hhis and improve patient adherence to Hhi treatment plans?**



**Key Points**
Treatment with sonidegib results in 10–15% fewer detectable AEs when compared with vismodegib.Treatment with sonidegib allows for excellent manageability of AEs with alternate-day dosing and dosage management capacity.Patients with Gorlin syndrome respond well to Hhis and are better able to tolerate adverse events.All of the side effects of Hhi therapy must be fully explained to the patient because an informed and motivated patient better tolerates side effects.Physical activity is important for managing muscle spasms; this is most effective if done immediately before going to sleep.For the control of dysgeusia, it is important to consult a dietician before starting therapy with Hhis.



**Question 3: When immunotherapy can be used as a second-line treatment after Hhis?**



**Key Point**
The use of the monoclonal antibody cemiplimab should be used only for patients that develop resistance under Hhi therapy



**Question 4: What are the educational and research needs in Hedgehog pathway inhibitors treatment?**



**Key Points**
Educational activities for General Practitioners, geriatrics and surgeons to increase awareness of the disease-promoting early diagnosis.Organise educational events through local medical organisations or local health administrations.


In conclusion, the Hhis, sonidegib and vismodegib, are increasingly being used as a valid medical therapy in managing laBCC. Several studies support the effectiveness and safety of these drugs. A panel of experts agrees on further potential uses of these drugs in addition to those already registered and recognised, especially in terms of the duration of therapy, use within a pulsed schedule, and use in adjuvant and neoadjuvant therapy. The panellists also agree on the need to conduct new studies that better confirm and outline these new potential uses.

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
