# Peer review of "Treatment of Advanced Basal Cell Carcinoma with Hedgehog Pathway Inhibitors: A Multidisciplinary Expert Meeting"

_cancers, 2021, doi:10.3390/cancers13225706_

Round 1
Reviewer 1 Report
In this report, the authors presented the opinion and experience in the treatment of local advanced basal cell carcinoma (laBCC), and provided information about new potential uses of Hhis sonidegib and vismodegib.
I have only some minor suggestions: the authors should further explain the difference between 'advanced basal cell carcinoma' and 'laBCC', it seemed that in the main text they discussed laBCC, and why the title stated 'advanced basal cell carcinoma'? In addition, the authors should check the format of text since it seems that some space are missing (such as Line 403).
Author Response
We thank the reviewer for his comments. The term "advanced basal cell carcinoma" includes "locally advanced basal cell carcinoma (laBCC)" and "metastatic basal cell carcinoma". In our text we deal in more detail with locally advanced basal cell carcinoma, also because it is much more frequent than metastatic. This point was highlighted more in the paper.
Reviewer 2 Report
In this paper authors describe the use of HhI for locally advanced BCC
There are a number of redundancy in the paper ( for exemple line 324 and line 336-324..; which are unnecessary
Line 57 the state that Sonidegib has a 6 time higher concentration in the skin than in the plasma which is true but the skin concentration of vismodegib was never measured so they cannot present it as an advantage. The same is repeated line 222.
Line 129 when they compare the two drugs I would be more cautious as it is side by side evaluation not face to face and evaluation method were very different.
page 167 about the neoadjuvant use of HHi the reference of Vismoneo study should be presented( eclinical medecine 2021 Bertrand et al)
Gorlin syndrome line 169: it should be mentionned that they all recurre after drug interruotion.
Additionnal the outcome of patients having achieved complete response and who stopped the treatment should be discussed ( Herms et al, JCO 2019)
page 324: greater immunogenicity of BCC during HHI therapy..... a reference is lacking.
Author Response
Reviewer 2
There are a number of redundancy in the paper ( for exemple line 324 and line 336-324..; which are unnecessary
Thank you for your comment that helps us improving our manuscript, the redundancy are removed in the revised manuscript.
Line 57 the state that Sonidegib has a 6 time higher concentration in the skin than in the plasma which is true but the skin concentration of vismodegib was never measured so they cannot present it as an advantage. The same is repeated line 222.
Thank you for your rightful comment. We have changed the manuscript to underline the fact that skin concentration of vismodegib was never measured. Please see lines 60-63 of the revised manuscript. In line 233 we deleted the sentence related to the skin concentration of vismodegib that it is never measured.
Line 129 when they compare the two drugs I would be more cautious as it is side by side evaluation not face to face and evaluation method were very different.
Thank you for your rightful suggestions. The reviewer is completely right, although it should be considered that the experts only discuss the BOLT study for sonidegib and the ERIVANCE study for vismodegib as they are pivotal trials with comparable design and end points.
page 167 about the neoadjuvant use of HHi the reference of Vismoneo study should be presented( eclinical medecine 2021 Bertrand et al)
Thank you for your suggestion, the reference of Vismoneo study is presented in the revised manuscript please see lines 172-175
Gorlin syndrome line 169: it should be mentionned that they all recurre after drug interruotion.
The reviewer is right, we have highlighted this aspect in the revised manuscript, please see lines: 182-186
Additionnal the outcome of patients having achieved complete response and who stopped the treatment should be discussed ( Herms et al, JCO 2019)
Thank you for you helpful suggestion. The outcome of patiente with complete response to vismodegib is now added to the revised manuscript, please see lines
page 324: greater immunogenicity of BCC during HHI therapy..... a reference is lacking.
A reference is now provided Please see lines 579-581 in the revised manuscript. Thank you for your suggestions
Reviewer 3 Report
This conference report summarizes the key findings of an expert panel meeting of Italian skin cancer specialists which was held in Florence in October 2020. As such, it examines important issues regarding patient care, treatment protocols and options.
It is mostly well written, and only requires several minor corrections.
Reference 10. typo: svismodegib
Lines 49-52: is mg/die the same as mg/day? Please change and make it uniform throughout the text
Define abbreviations at first mention (ORR, AE, RT)
Lines 169-176: make the nomenclature of the Gorlin-Goltz (Gorlin) syndrome uniform throughout the text
Lines 189-192: please provide a reference for this statement
Lines 193-196: please provide a reference for this statement
Lines 265 and 270: here you refer to ageusia, while the rest of the report mentions dysgeusia. Make it uniform or explain why ageusia is mentioned in this specific instance
Please make the terminology and abbreviations uniform: HHI and Hhi are both used in the text, but HHI abbreviation is used mostly for Question 3. Please change it to match the rest od the text.
Line 290: please provide a reference for UV-induced pathogenesis of BCC
Line 303: please check the term HPI therapy
Line 360: typo: furthere
Where are the Tables 1 and 2 referred to in lines 380-381 and 395-396? Does this refer to the key points? If so, these are not tables and should not be listed as such. Also, why are key points nicely summarized for Questions 1 and 2, but Questions 3 and 4 are overlooked? I would suggest incorporating the key points for each Question into the conclusion section, as these are the take home messages from this meeting.
Author Response
Reviewer 3
This conference report summarizes the key findings of an expert panel meeting of Italian skin cancer specialists which was held in Florence in October 2020. As such, it examines important issues regarding patient care, treatment protocols and options.
It is mostly well written, and only requires several minor corrections.
Thank you for your positive comments.
Reference 10. typo: svismodegib
Thank you for your suggestion, the word you mention in reference 10 is now corrected.
Lines 49-52: is mg/die the same as mg/day? Please change and make it uniform throughout the text
The term mg/die is now replaced by mg/day in the revised text. Thank you for your right suggestion.
Define abbreviations at first mention (ORR, AE, RT)
Thank you for your rightful suggestion. We have checked the text to define all the abbreviations at first mention
Lines 169-176: make the nomenclature of the Gorlin-Goltz (Gorlin) syndrome uniform throughout the text
The nomenclature of Gorlin-Goltz syndrome is now uniform throughout the text
Lines 189-192: please provide a reference for this statement
A reference is now provided. Please see lines 548-549 in the revised manuscript. Thank you for your suggestions
Lines 193-196: please provide a reference for this statement
A reference is now provided Please see lines 550-552 in the revised manuscript. Thank you for your suggestions
Lines 265 and 270: here you refer to ageusia, while the rest of the report mentions dysgeusia. Make it uniform or explain why ageusia is mentioned in this specific instance
Thank you for your suggestion: the term dysgeusia is now uniformed in the text.
Please make the terminology and abbreviations uniform: HHI and Hhi are both used in the text, but HHI abbreviation is used mostly for Question 3. Please change it to match the rest od the text.
The terminology and abbreviations for Hhi is now standardise in the text. Thank you for your suggestion
Line 290: please provide a reference for UV-induced pathogenesis of BCC
A reference is now provided Please see lines 570-571 in the revised manuscript. Thank you for your suggestions
Line 303: please check the term HPI therapy
We apologize for this typing error, the word is changed in the revised manuscript.
Line 360: typo: furthere
We apologize for this typing error, the word is now corrected in the revised manuscript.
Where are the Tables 1 and 2 referred to in lines 380-381 and 395-396? Does this refer to the key points? If so, these are not tables and should not be listed as such. Also, why are key points nicely summarized for Questions 1 and 2, but Questions 3 and 4 are overlooked? I would suggest incorporating the key points for each Question into the conclusion section, as these are the take home messages from this meeting.
The key points are now summarized for each of the four questions and they are include in the conclusion section. Thank you for your rightful suggestion.
Round 2
Reviewer 2 Report
The paper has been improved according to the reveiwer's comment